# Association of MG53 with presence of type 2 diabetes mellitus, glycemic control, and diabetic complications

**Burak Andaç** [1] *, **Eray Özgün**[2], **Buket Yılmaz Bülbül**[1], **Serpil Yanık Çolak**[1], **Mine Okur**[1], **Ali Cem Yekdeş** [3], **Eftal Öcal**[4], **Mehmet Emin Tapan**[4], **Mehmet Çelik**[1]

1 Department of Endocrinology and Metabolism, Medical Faculty, Trakya University, Edirne, Turkey, 2 Department of Biochemistry, Medical Faculty, Trakya University, Edirne, Turkey, 3 Department of Public Health, Medical Faculty, Trakya University, Edirne, Turkey, 4 Department of Internal Medicine, Medical Faculty, Trakya University, Edirne, Turkey

* drburakandac87@gmail.com

**Data Availability Statement:** All relevant data are within the Supporting Information files.

**Funding:** Financial support was provided for our project numbered 2022/13 by Trakya University

## Abstract

### Objectives

Mitsugumin 53 (MG53) is a myokine that acts as a membrane repair protein in tissues. Data on the effect of MG53 on insulin signaling and type 2 diabetes mellitus (T2 DM) are still unknown; most are from preclinical studies. Nevertheless, some researchers have argued that it may be a new pathogenic factor, and therapies targeting MG53 may be a new avenue for T2 DM. Our study aims to evaluate the relationship of circulating MG53 levels with the presence of diabetes, diabetic complications, and glycemic control.

### Methods

We conducted a case-control study with 107 patients with T2 DM and 105 subjects without insulin resistance-related disease. Concurrent blood samples were used for serum MG53 levels and other biochemical laboratory data. MG53 concentration was measured using Human-MG53, an enzyme-linked immunosorbent assay kit (Cat# CSB-EL024511HU).

### Results

We found no difference in MG53 levels between the diabetic and control groups (p = 0.914). Furthermore, when the subjects were divided into tertiles according to their MG53 levels, we did not find any difference between the groups in terms of the presence of diabetes (p = 0.981). Additionally, no correlation was observed between weight, BMI, waist circumference, systolic and diastolic blood pressure, fasting blood glucose, HbA1c, albumin excretion in the urine, e-GFR levels, and MG53. Finally, MG53 levels were similar between the groups with and without microvascular and macrovascular complications of diabetes.

### Conclusion

Our research finding provides insightful clinical evidence of lack of association between the levels of MG53 and T2 DM or glycemic control, at least in the studied population of Turkeys

Scientific Research Projects Unit. The funders had no role in study design, data collection and analysis, decision to publish, or preparation of the manuscript.

**Competing interests:** The authors have declared that no competing interests exist.

ethnicity. However, further clinical studies are warranted to establish solid evidence of the link between MG53, insulin resistance and glycemic control in a wider population elsewhere in the world.

## Introduction

Type 2 diabetes mellitus (T2DM) is estimated to influence more than 400 million people worldwide [1]. Furthermore, the incidence of diabetes is predicted to increase and influence around one in three people by the year 2050 [2]. Considering its chronic complications and mortality rates, research on the pathophysiology and treatment of T2 DM is also increasing. Myokines related to insulin resistance at the muscle level have been the subject of various studies in terms of diabetes pathophysiology [3].

One of these myokines, mitsugumin 53 (MG53), also known as TRIM72, is a multifunctional protein that belongs to the tripartite motif (TRIM) family and is abundantly expressed in skeletal and cardiac muscle [4]. Besides its essential physiological roles, MG53 has also been shown to act as an important pathogenic factor in various diseases [5]. For example, MG53 maintains cardiac and skeletal muscle integrity by participating in cell membrane repair of the heart, skeletal muscle, and other tissues [6, 7]. Acute elevation of intracellular MG53 also has protective effects against ischemia/reperfusion injury of the myocardium [8].

Although its important function as a membrane repair protein has been elucidated [9], the role of MG53 in many metabolic processes, especially in the insulin signaling pathway, is controversial. Although some study results from preclinical studies in animal models have suggested that MG53 upregulation may lead to metabolic diseases such as T2 DM and obesity by causing insulin resistance in skeletal muscle [10, 11], there are also studies with opposite results. Despite the hypothesis that MG53 elevation may be a pathogenic factor in T2 DM [10], many studies have not established a causal relationship between insulin resistance and MG53 [12–14].

Consequently, MG53 has been shown to have both beneficial and negative effects on many diseases. To date, almost all studies of MG53's association with insulin sensitivity and DM have been performed in animal models. Although preclinical studies are conflicting, MG53 has been suggested as a novel pathogenic factor for diabetes in animal models. Observing its relationship with diabetes, diabetic complications, and glycemic control in human studies may open a new avenue for treating type 2 diabetes and its complications. In our study, we aimed to examine whether there is a difference in serum MG53 levels between the patient group diagnosed with type 2 DM and the healthy control group without metabolic syndrome and diabetes and to determine the relationship between diabetic complications and glycemic control and MG53 levels in the patient group.

## Materials and methods

This single-center, case-control study was conducted between March 2022 and December 2022. The Ethics Committee of Trakya University approval was granted before the study (TUTF-BAEK 2021/414). The study was conducted in accordance with the Declaration of Helsinki about the rules of ethics in medical research. Individuals who met the inclusion criteria and obtained written informed consent were included in the study.

The effect size was calculated as 0.5 based on the study performed by Hongyang Xie et al. [15]. Considering the effect size, we included 212 subjects (107 patients with T2 DM and 105 healthy individuals) with a 5% significance level and 95% power. One hundred seven patients

who were followed up with the diagnosis of T2 DM at our outpatient clinic and who agreed to participate in the study were randomly selected and included. Patients without known chronic disease were included in the control group.

- *Exclusion criteria for the control group*

- Obesity (BMI$\geq$30 kg/m$^2$)

- Prediabetes or metabolic syndrome

- Hypertension (HT)

- Atherosclerotic cardiovascular disease

- Type 1 DM

- Acute renal failure

- Chronic liver disease

In addition, pregnant women-breastfeeding individuals, those with severe acute illnesses or using glucocorticoids, and those with conditions that cause muscle destruction (early trauma, early post-MI, rheumatological diseases such as polymyositis, and neurological diseases such as myasthenia gravis) were excluded from the both study groups.

The presence of HT was defined as: Patients with an average office blood pressure measurement $\geq$140/$\geq$90 mmHg, an average home blood pressure follow-up of $\geq$130/$\geq$80 mmHg, or those using anti-hypertensive medication. The blood pressure values used in the correlation analysis were determined according to the average of the 10-day home monitoring. Diabetic microvascular and macrovascular complication data of the patient group were recorded from file records and hospital information systems. The presence of at least one of macular edema, microaneurysms, hard exudates, hemorrhages, intraretinal microvascular abnormalities, and pathological preretinal neovascularization findings as a result of fundus evaluation was evaluated as retinopathy. In a 3–6 month period, abnormal albumin excretion ($\geq$30 mg/day) of at least two 24-hour urine sample examinations or persistent low e-GFR ($<$60 ml/min/1.73 m$^2$, CKD-EPI formula) was defined as nephropathy (in patients without nondiabetic causes of proteinuria and low e-GFR, such as febrile illness, urinary tract infection, acute renal failure, and severe heart failure). Diabetic polyneuropathy was diagnosed based on a combination of typical symptoms and signs, as summarized in the 2017 position statement by American Diabetes Association [16]. The presence of macrovascular complications was defined as at least one of the following conditions: acute coronary syndrome, history of myocardial infarction, stable or unstable angina, coronary or other arterial revascularization, stroke, transient ischemic attack, or peripheral arterial disease. Obesity was defined as a BMI of $\geq$30 kg/m$^2$.

Concurrent blood samples were used for serum MG53 levels and other biochemical laboratory data. HbA1c measurement was performed by using Tosoh G8 high-performance liquid chromatography analyzer with its original solutions. Blood samples were centrifuged at 1000 g for 15 minutes and serum was separated. Serum total cholesterol, LDL-cholesterol, HDL-cholesterol, triglyceride, ALT, AST, and creatinine levels were measured by Roche C702 chemistry analyzer using its original kits. *After separation, s*erum samples were also immediately frozen at -80˚C for use later to measure MG53 levels. Serum MG53 levels were measured using a commercial Human-MG53 enzyme-linked immunosorbent assay (ELISA) kit (Cat# CSB-EL024511HU) according to the kit's original protocol. Intra-assay %CV of the kit was $<$8% and we calculated the limit of detection of the kit as 8.57 pg/mL by measuring the ten replicates of zero calibrator (sample diluent). All samples were measured blinded to the patient data.

Subjects with MG53 levels below the detection limit were excluded from the statistical evaluation. Statistical evaluation was continued with the remaining subjects. Continuous variables with normal distribution were expressed as mean ± standard deviation. Median (25th-75th percentiles) were used for variables assumed non-normal distribution. Categorical variables were stated as number (n) and percentage (%).The Shapiro-Wilk test was used to assess the distribution for all variables. The comparison between the study groups was evaluated using the independent sample t test for quantitative data analysis with normal distribution. The comparison between study groups was evaluated using the Mann-Whitney U test for quantitative data analysis with non-normal distribution. Pearson Chi-square analysis was used to compare the groups for qualitative data analysis. The correlation between variables was evaluated with the Spearman's rank correlation test. P values of <0.05 were regarded as statistically significant. Statistical analyses were carried out using SPSS 22.0 version (IBM Corporation, Armonk, NY, USA).

## Results

The main clinical and anthropometric characteristics of the T2 DM group and control subjects are reported and compared in Table 1. The mean age of all subjects was 46.76 ± 10.90 years. Of all participants, 86 (40.6%) were male, and 126 (59.4%) were female. The two groups were matched for age and sex. MG53 levels (including those below the detection limit) were divided into tertiles as low, medium, and high. The presence of diabetes mellitus in three groups was evaluated by cross-tabulation. Accordingly, 51.4% of the participants in the low group, 50% in the middle group, and 50% in the high group were found to have DM. No significant difference was observed between the groups (p:0.981). We found the limit of detection 8.57 pg/ml for the MG53 kit. The serum MG53 levels of 36 patients in the DM group and 34 subjects in the control group were lower than the detection limit (<8.57 pg/mL). In analyzes after excluding those below the detection limit, there was no difference in MG53 levels between the diabetic and control groups (p = 0.914) (Table 2). Likewise, in the patient group with T2 DM, MG53 levels were similar between those with and without obesity (p = 0.271).

The results of the correlation analysis between MG53 serum levels and clinical and laboratory parameters are shown in Table 3. Circulating MG53 levels were not affected by age or sex. In addition, no correlation was found between weight, BMI, waist circumference, waist/hip ratio, systolic blood pressure (SBP), diastolic blood pressure (DBP), fasting blood glucose (FBG), HbA1c, albumin excretion in the urine, e-GFR levels and MG53. Only a very weak-weak correlation was found with serum lipid levels (Table 3).

In addition, MG53 median levels were similar between the groups with and without microvascular and macrovascular complications of diabetes (Table 4).

## Discussion

Insulin resistance is an essential mechanism in the formation of T2DM. Previous research on insulin resistance has revealed the roles of adipose tissue and liver [17]. However, recent studies have shown that skeletal muscle insulin resistance may be the primary and central pathological process during the development of global metabolic disorders [10, 18]. Therefore, the role of myokines secreted from skeletal muscle in the pathogenesis of insulin resistance and T2 DM or the relationship between metabolic diseases and the expression status of myokines in muscle has formed the background of some studies. One of these myokines, MG53, has been investigated for the development of insulin resistance and diabetes, especially in animal models. However, limited human studies on this subject have been conducted with relatively few patients.

**Table 1. Comparison of main characteristics in DM and control groups.**

| Parameters | | Control Subjects (n:105) Median (IQR) | Diabetic Individuals (n:107) Median (IQR) | p |
|---|---|---|---|---|
| Age (years) | | 46,30±10,84* | 47,21±11,00* | 0,549[a] |
| Sex | Male | 41 (%39,0)** | 45 (%42,1)** | 0,656[b] |
| | Female | 64 (%61,0)** | 62 (%57,9)** | |
| Height (cm) | | 165,00(160,00–172,50) | 166,00(160,00–173,00) | 0,906[c] |
| Weight (kg) | | 65,00(60,00–75,00) | 89,00(79,00–105,00) | 0,001[c] |
| BMI (kg/m$^2$) | | 24,00(22,13–26,36) | 32,30(28,30–36,49) | 0,001[c] |
| Waist circumference (cm) | | 85,00 (77,50–92,00) | 110,00(102,00–120,00) | 0,001[c] |
| Waist/Hip ratio | | 0,84±0,07* | 0,94±0,08* | 0,001[a] |
| Presence of Obesity | Obesity (+) | - | 73 (%68,2)** | 0,001[b] |
| | Obesity (-) | 105 (%100,0)** | 34 (%31,8)** | |
| SBP (mmHg) | | 110,00(100,00–120,00) | 120,00(115,00–130,00) | 0,001[c] |
| DBP (mmHg) | | 70,00(65,00–75,00) | 75,00(70,00–80,00) | 0,001[c] |
| FBG (mg/dL) | | 90,00(84,00–94,00) | 137,00(112,00–180,00) | 0,001[c] |
| HbA1c (%) | | 5,60(5,40–5,90) | 7,90(6,80–10,00) | 0,001[c] |
| Triglyceride (mg/dL) | | 101,00(73,00–158,00) | 155,00(110,00–217,00) | 0,001[c] |
| Total Cholesterol (mg/dL) | | 195,00(169,00–224,00) | 178,00(153,00–217,00) | 0,033[c] |
| LDL-C(mg/dL) | | 124,00(101,50–148,00) | 108,00(92,00–139,00) | 0,015[c] |
| HDL-C (mg/dL) | | 51,00(41,00–62,00) | 41,00(36,00–49,00) | 0,001[c] |
| Non-HDL-C (mg/dL) | | 141,00(112,00–170,50) | 137,00(106,00–166,00) | 0,361[c] |
| Creatinine (mg/dL) | | 0,75(0,65–0,91) | 0,73(0,63–0,90) | 0,481[c] |
| e-GFR (mL/min/1.73 m$^2$) | | 101,00(89,50–109,00) | 103,00(92,00–110,00) | 0,710[c] |
| ALT (U/L) | | 15,00(12,00–21,00) | 21,00(15,00–31,00) | 0,001[c] |
| AST (U/L) | | 17,00(15,00–22,00) | 19,00(15,00–23,00) | 0,233[c] |

**SBP:**Systolic Blood Pressure; **DBP:**Diastolic Blood Pressure; **FBG:**Fasting Blood Glucose; **HbA1c:** Glycated hemoglobin; **LDL-C:**Low Density Lipoprotein-Cholesterol; **HDL-C**: High Density Lipoprotein-Cholesterol; **e-GFR:** Estimated Glomerular Filtration Rate; **ALT**: Alanine Aminotransferase; **AST:** Aspartate Aminotransferase

*Mean±SD

**n(%)

[a] Independent samples t test

[b] Chi-square test

[c] Mann-Whitney U test

The results from these studies are contradictory. Philouze et al. [13] showed that MG53 gene knockdown in muscle cells did not impair insulin sensitivity, and recombinant human MG53 did not cause a change in insulin response in skeletal muscle cells. Moreover, Wang et al. [12] found no effect on glucose handling in db/db mice produced by sustained elevation or whole-body ablation of MG53 in the bloodstream. These findings align with the research of Ma et al. [19], who achieved similar in vivo and in vitro results using alternative routes. Contrary to these studies, in a study by Wu et al. [20], it was shown that intravenously administered recombinant MG53 protein inhibited insulin response in many organs. In contrast, the administration of monoclonal antibodies that neutralize serum MG53 protein has been shown to improve hyperglycemia and increase insulin sensitivity in diabetic mice. Regarding its possible role in the pathogenesis of T2 DM, a study [10] suggested that increased MG53 expression in muscle may cause insulin resistance through E3-ligase-mediated degradation of insulin receptor substrate 1 (IRS-1); however, two separate studies [21, 22] argued that IRS-2, IRS-3, and IRS-4 also contribute to insulin signaling and that disruption of IRS-1 alone is insufficient to trigger the development of T2 DM. As a result, these studies suggest skepticism about the role of MG53 in regulating insulin signaling. Even though we cannot evaluate causality due to

**Table 2. Comparison of main characteristics in DM and control groups (MG53<8.57 pg/ml excluded).**

| Parameters | | Control Subjects (n:71) Median (IQR) | Diabetic Individuals (n:71) Median (IQR) | p |
|---|---|---|---|---|
| Age (years) | | 47,04±9,83* | 48,06±10,29* | 0,512[a] |
| Sex | Male | 25(%35,21)** | 30(%42,25)** | 0,491[b] |
| | Female | 46(%64,79)** | 41(%57,75)** | |
| Height (cm) | | 165(158–172) | 166(160–173) | 0,555[c] |
| Weight (kg) | | 64(59,5–73,5) | 90(78–103) | <0,001[c] |
| BMI (kg/m$^2$) | | 24(22,14–26,20) | 32,7(27,9–36,3) | <0,001[c] |
| Waist circumference (cm) | | 85(77,5–90) | 110(101–120) | <0,001[c] |
| Waist/Hip ratio | | 0,84±0,08* | 0,94±0,09* | 0,445[a] |
| Presence of Obesity | Obesity (+) | - | 47(%66,19)** | <0,001[b] |
| | Obesity (-) | 71(%100)** | 24(%33,81)** | |
| SBP (mmHg) | | 110(100–120) | 120(115–130) | <0,001[c] |
| DBP (mmHg) | | 70(65–75) | 75(70–80) | <0,001[c] |
| FBG (mg/dL) | | 90(84–93) | 144(112,5–201,5) | <0,001[c] |
| HbA1c (%) | | 5,6(5,4–5,9) | 8,1(6,8–10,3) | <0,001[c] |
| Triglyceride (mg/dL) | | 105(77–164) | 164(120,5–231) | <0,001[c] |
| Total Cholesterol (mg/dL) | | 199(169–229) | 185(160,5–223) | 0,170[c] |
| LDL-C(mg/dL) | | 124(106–149) | 113(95–146,5) | 0,134[c] |
| HDL-C (mg/dL) | | 54(42–59) | 41(36–48) | <0,001[c] |
| Non-HDL-C (mg/dL) | | 141(120–172) | 143(114,5–170) | 0,744[c] |
| Creatinine (mg/dL) | | 0,72(0,64–0,84) | 0,72(0,62–0,89) | 0,673[c] |
| e-GFR (mL/min/1.73 m$^2$) | | 102,9(91–109,5) | 103(94,3–109,5) | 0,744[c] |
| ALT (U/L) | | 16(13–21) | 21(15–32,5) | 0,001[c] |
| AST (U/L) | | 18(15–22) | 18(15–23,5) | 0,671[c] |
| MG53(<8.57 pg/ml values excluded) | | 61,43(30–138,57) | 64,29(34,29–122,86) | 0,914[c] |

**SBP:**Systolic Blood Pressure; **DBP:**Diastolic Blood Pressure; **FBG:**Fasting Blood Glucose; **HbA1c:** Glycated hemoglobin; **LDL-C:**Low Density Lipoprotein-Cholesterol; **HDL-C:** High Density Lipoprotein-Cholesterol; **e-GFR:** Estimated Glomerular Filtration Rate; **ALT:** Alanine Aminotransferase; **AST:** Aspartate Aminotransferase

*Mean±SD

**n(%)

[a] Independent samples t test

[b] Chi-square test

[c] Mann-Whitney U test

the study design, we showed no relationship between clinical and laboratory parameters associated with insulin resistance (BMI, waist circumference, waist/hip ratio, arterial blood pressure) and MG53 serum levels.

In addition to the lack of causality evidence between Mg53 and DM, the results of injections of MG53 antibodies to lower blood glucose levels have also been criticized by some authors. Wu et al. [20] found a decrease in glucose from ∼425 to 375 mg/dl with MG53 antibody injection. However, some authors have found standard errors in assessing blood glucose levels in db/db mice [23, 24]. In addition, the injected dose of MG53 antibody was at a concentration >10$^6$-fold higher than the MG53 levels detected in the blood [23]. Therefore, the safety of high antibody therapy doses regarding possible adverse effects on essential organ functions is questionable. In addition, blood glucose levels of ∼375 mg/dl achieved with treatment are still high in terms of the development of complications of hyperglycemia.

Although several published manuscripts have reported an increase in MG53 in diabetic animal models [10, 20, 25], many independent investigators have determined MG53 levels in

**Table 3. Correlation analyzes of the parameters with MG53 levels (<8.57 pg/ml excluded).**

| Parameters | | Diabetic Group MG53 pg/ml | Control Group MG53 pg/ml | All Subjects MG53 pg/ml |
|---|---|---|---|---|
| Age (years) | r | -0,041 | 0,181 | 0,065 |
| | p | 0,734 | 0,132 | 0,441 |
| Height (cm) | r | 0,018 | -0,059 | -0,017 |
| | p | 0,882 | 0,626 | 0,837 |
| Weight (kg) | r | 0,039 | 0,052 | 0,053 |
| | p | 0,748 | 0,667 | 0,532 |
| BMI (kg/m$^2$) | r | 0,167 | 0,175 | 0,127 |
| | p | 0,165 | 0,143 | 0,132 |
| Waist circumference (cm) | r | 0,200 | 0,083 | 0,098 |
| | p | 0,094 | 0,489 | 0,244 |
| Waist/Hip ratio | r | -0,005 | 0,168 | 0,088 |
| | p | 0,967 | 0,162 | 0,299 |
| SBP (mmHg) | r | 0,111 | -0,024 | 0,025 |
| | p | 0,358 | 0,843 | 0,764 |
| DBP (mmHg) | r | 0,139 | 0,049 | 0,089 |
| | p | 0,247 | 0,686 | 0,292 |
| FBG (mg/dL) | r | -0,057 | 0,168 | 0,015 |
| | p | 0,637 | 0,162 | 0,862 |
| HbA1c (%) | r | 0,082 | - | 0,082 |
| | p | 0,496 | | 0,496 |
| Triglyceride (mg/dL) | r | 0,321 | 0,307 | 0,309 |
| | p | 0,006 | 0,009 | 0,001 |
| Total Cholesterol (mg/dL) | r | 0,338 | 0,167 | 0,246 |
| | p | 0,004 | 0,164 | 0,003 |
| LDL-C (mg/dL) | r | 0,287 | 0,105 | 0,199 |
| | p | 0,015 | 0,385 | 0,017 |
| HDL-C (mg/dL) | r | -0,189 | -0,232 | -0,2227 |
| | p | 0,114 | 0,051 | 0,008 |
| Non-HDL-C (mg/dL) | r | 0,392 | 0,339 | 0,277 |
| | p | 0,001 | 0,004 | 0,001 |
| Creatinine (mg/dL) | r | -0,010 | 0,128 | 0,066 |
| | p | 0,934 | 0,287 | 0,438 |
| e-GFR (mL/min/1.73 m$^2$) | r | -0,019 | -0,704 | -0,072 |
| | p | 0,874 | 0,001 | ,0,397 |
| ALT (U/L) | r | -0,107 | 0,280 | 0,071 |
| | p | 0,375 | 0,018 | 0,398 |
| AST (U/L) | r | 0,005 | 0,328 | 0,042 |
| | p | 0,964 | 0,005 | 0,621 |
| Creatinine clearance (mL/min) (24 h urine) | r | 0,074 | - | - |
| | p | 0,565 | | |
| Microalbuminuria | r | -0,011 | - | - |
| | p | 0,931 | | |
| DM duration (in years) | r | 0,029 | - | - |
| | p | 0,810 | | |

SBP:Systolic Blood Pressure; **DBP:**Diastolic Blood Pressure; **FBG:**Fasting Blood Glucose; **HbA1c:** Glycated hemoglobin; **LDL-C:**Low Density Lipoprotein-Cholesterol; **HDL-C:** High Density Lipoprotein-Cholesterol; **e-GFR:** Estimated Glomerular Filtration Rate; **ALT:** Alanine Aminotransferase; **AST:** Aspartate Aminotransferase **r:**Spearman correlation coefficient.

**Table 4. Distribution of MG53 levels in various groups (<8.57 pg/ml excluded).**

|  |  | MG53 pg/ml Median (IQR) | p |
|---|---|---|---|
| **Hypertension** | (-) (n:35) | 57,14(32,14–95,71) | 0,959[a] |
|  | (+) (n:36) | 60,71(31,43–133,93) |  |
| **Obesity** | (-) (n:24) | 48,57(20,71–90,00) | 0,271[a] |
|  | (+) (n:47) | 64,29(36,42–128,57) |  |
| **Retinopathy** | (-) (n:58) | 57,14(35,00–122,86) | 0,924[a] |
|  | (+) (n:10) | 63,57(23,57–140,35) |  |
| **Nephropathy** | (-) (n:56) | 57,14(34,29–122,86) | 0,834[a] |
|  | (+) (n:12) | 66,43(27,14–118,56) |  |
| **Neuropathy** | (-) (n:56) | 62,14(33,93–123,57) | 0,468[a] |
|  | (+) (n:15) | 42,86(24,28–102,85) |  |
| **At least one microvascular complication** | (-) (n:46) | 58,57(34,64–122,86) | 0,843[a] |
|  | (+) (n:25) | 52,86(27,14–125,71) |  |
| **Macrovascular complication** | (-) (n:59) | 57,14(30,00–120,00) | 0,504[a] |
|  | (+) (n:12) | 62,85(35,71–133,93) |  |

[a]Mann-Whitney U test.

muscle and serum samples from diabetic animal and human models to be at or below the same level as non-diabetic [12–14, 19, 24]. Our study showed no difference in serum MG53 levels between diabetic patients and control subjects. In addition, we did not detect a correlation between serum MG53 levels and fasting serum glucose and HbA1c levels. Wang et al. [12] found no difference in MG53 levels between diabetic people and animals and the control group, similar to our study. They also did not find any correlation with FBG levels. In contrast, Wu et al. [20]reported that serum MG53 levels were elevated in humans with T2 DM. In addition, the researchers found a positive correlation between FBG and MG53 levels. In the two studies mentioned, Wang et al. [12] used the same commercial antibody used by Wu et al. [20], moreover, these antibodies recognized nonspecific bands in serum from multiple mg53-/- mouse strains. In that case, care should be taken when interpreting the amount and identity of MG53 in the serum. In another study, MG53 was significantly lower in the control group than in patients with T2DM; however, there was no difference between individuals with impaired glucose tolerance and T2 DM subjects [26]. Moreover, MG53 was divided into quartiles in this study; there was no difference between the fourth and first quartiles regarding BMI, waist circumference, systolic and diastolic blood pressure, and HbA1c levels. These results are compatible with our study in terms of not establishing a relationship between serum MG53 levels and obesity, HT, and glycemic control. In particular, the inability to show the relationship between HbA1c and MG53 serum levels may make treatment approaches aiming to reduce MG53 debatable.

In a study of diabetic retinopathy, MG53 overexpression was found to attenuate high glucose-induced dysfunction in human retinal microvascular endothelial cells [27]. In addition, MG53 has been shown to reduce neuroinflammation and neurotoxicity [28]. Furthermore, other researchers concluded that MG53 might reduce renal fibrosis and intrarenal inflammation by inhibiting cytokine-induced activation of NF-κB [29]. Our study found no difference in circulating MG53 levels between groups with and without microvascular complications in people with diabetes. In addition, MG53 serum levels were similar in the group with macrovascular complications compared to those without. The relatively small number of patients with diabetic complications and the fact that we did not evaluate MG53 levels at the tissue level

limit the predictions of our study on this subject. However, when evaluated together with the preclinical studies mentioned above, it can be thought that possible treatments aiming to decrease serum MG53 levels will not positively affect the micro and macrovascular complications of diabetes. Given that MG53 is a cell membrane repair protein, it is not surprising that MG53 upregulation is involved in repairing damage to endothelial cells. That is another circumstance that makes the goal of lowering serum MG53 levels doubtful.

Variations in serum MG53 levels may be due to the fact that MG53 is an intracellular myokine. The fact that we found serum MG53 levels below the detection limit of approximately one-third of the individuals in both study groups may be due to the intracellular nature of this molecule. Furthermore, factors such as exercise or muscle damage can affect MG53 release. It may also have an increasing or decreasing circadian rhythm during the day; however, we did not find such a study in the literature. In addition, the relatively short half-life of MG53 in the bloodstream may be one of the reasons for these variations [30]. Similar variations were also demonstrated by Wu et al. [20], indicating that it would be difficult to establish a relationship between serum levels of MG53 and metabolic diseases.

When interpreting the results of our study, it should be considered that it was single-centered and performed only in the Turkish ethnic group. There may be an association between MG53 and insulin resistance and diabetes in other ethnic populations elsewhere. Furthermore, given the high prevalence of insulin resistance in Turkey and the lack of comprehensive exclusion of insulin resistance in the studied population, it is challenging to generalize the results of this study. Moreover, there are limited clinical studies and, in particular, multicenter clinical randomized trials or systematic reviews to build solid evidence for this conclusion. The imperative finding of this study cannot be generalized to other populations because the study methods (design, setting, and population) do not meet external validity. The design of our study cannot refute previous evidence but rather provides clinical insight.

## Conclusion

Our research has shown that there is no significant connection between the levels of circulating MG53 and T2 DM or glycemic control in the studied population of Turkish ethnicity. However, we believe that further clinical studies should be conducted to establish a more solid link between MG53, insulin resistance, and glycemic control across a broader range of populations worldwide.

### Study limitations

There are some limitations in our study that should be considered. First, since MG53 is an intracellular protein secreted from skeletal and cardiac muscle, subjects' exercise habits may affect serum levels. There is no data related to exercise habits in our study. However, subjects were selected from those who had not exercised heavily in the past three days, which could have caused muscle damage. Furthermore, it should also be kept in mind that oral antidiabetic and insulin therapy may affect MG53 levels. Finally, the control group consisted of healthy individuals who did not meet the clinical and demographic parameters associated with insulin resistance. However, the lack of HOMA-IR index data may suggest that the presence of insulin resistance in the control group could not be excluded entirely.

## Supporting information

**S1 Dataset.**
(SAV)

## Acknowledgments

We would like to express our deep and sincere gratitude to our research supervisor, Dr. S. Guldiken, Endocrinology and Metabolism.

## Author Contributions

**Conceptualization:** Burak Andaç, Eray Özgün, Buket Yılmaz Bülbül, Serpil Yanık Çolak, Mine Okur, Ali Cem Yekdeş, Eftal Öcal, Mehmet Emin Tapan, Mehmet Çelik.

**Data curation:** Burak Andaç, Serpil Yanık Çolak, Eftal Öcal, Mehmet Emin Tapan.

**Formal analysis:** Burak Andaç, Eray Özgün, Buket Yılmaz Bülbül, Serpil Yanık Çolak, Mine Okur, Ali Cem Yekdeş, Mehmet Çelik.

**Funding acquisition:** Mehmet Çelik.

**Methodology:** Burak Andaç, Eray Özgün, Buket Yılmaz Bülbül, Serpil Yanık Çolak, Mine Okur, Mehmet Çelik.

**Resources:** Eftal Öcal, Mehmet Emin Tapan.

**Supervision:** Eray Özgün, Buket Yılmaz Bülbül, Ali Cem Yekdeş, Mehmet Çelik.

**Writing – original draft:** Burak Andaç, Eray Özgün, Mehmet Çelik.

**Writing – review & editing:** Burak Andaç, Eray Özgün, Buket Yılmaz Bülbül, Serpil Yanık Çolak, Mine Okur, Ali Cem Yekdeş, Eftal Öcal, Mehmet Emin Tapan, Mehmet Çelik.

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
