## [Decision Letter · Decision Letter 0]

14 Jun 2023

PONE-D-23-16760Association of MG53 with presence of type 2 diabetes mellitus, glycemic control, and diabetic complicationsPLOS ONE

Dear Dr. ANDAÇ,

Thank you for submitting your manuscript to PLOS ONE. After careful consideration, we feel that it has merit but does not fully meet PLOS ONE’s publication criteria as it currently stands. Therefore, we invite you to submit a revised version of the manuscript that addresses the points raised during the review process.

We look forward to receiving your revised manuscript.

Kind regards,

Donovan Anthony McGrowder, PhD., MA., MSc

Academic Editor

PLOS ONE

Journal Requirements:

"We would like to express our deep and sincere gratitude to our research supervisor, Dr. S. Guldiken, Endocrinology and Metabolism. We are also grateful to Trakya University Scientific Research Projects Unit for its financial support."

"Financial support was provided for our project numbered 2022/13 by Trakya University Scientific Research Projects Unit. The funders had no role in study design, data collection and analysis, decision to publish, or preparation of the manuscript."

Reviewers' comments:

Reviewer's Responses to Questions

**Comments to the Author**

1. Is the manuscript technically sound, and do the data support the conclusions?

Reviewer #1: Partly

Reviewer #2: Yes

2. Has the statistical analysis been performed appropriately and rigorously? 

Reviewer #1: Yes

Reviewer #2: Yes

3. Have the authors made all data underlying the findings in their manuscript fully available?

Reviewer #1: Yes

Reviewer #2: Yes

4. Is the manuscript presented in an intelligible fashion and written in standard English?

Reviewer #1: Yes

Reviewer #2: Yes

5. Review Comments to the Author

Reviewer #1: General comments

Diabetes mellitus and its associated complications have a significant negative impact on the health and socioeconomic status of a given population. Therefore, it is imperative to keep on looking for further supplement strategies e.g. novel treatments to avert its incidence and burden across the world.

The manuscript is well-prepared and written in standard English. The introduction has a logical flow of ideas. The authors used a good design, and the results reflected the study objectives.

On the other hand, there are some issues of concern that the authors should address to improve their current manuscript.

Specific comments

Study conclusion:

The imperative finding of this study cannot be generalized to other populations because the study methods (design, setting and population) do not meet external validity. Moreover, given the high prevalence of insulin resistance in Turkey and the lack of exhaustive exclusion of insulin resistance in the studied population then it is difficult to generalize the finding of this study.

The study design, setting (single centre) and population cannot refute the previous evidence rather it provides a contrary clinical insight. Furthermore, it might be that there is an association between MG53 and insulin resistance and diabetes in other ethnic populations elsewhere. Moreover, there are limited clinical studies and in particular, multicenter clinical randomized trials and/or systematic reviews to build solid evidence for this conclusion. Therefore, it is appropriate to make a conclusion based on the internal validity of the study and the studied population.

In this regard, I suggest the author revise and rephrase the conclusion both in the abstract and discussion sections. For example, in the abstract, the conclusion can be read as “Our research finding provides insightful clinical evidence of lack of association between the levels of MG53 and T2 DM or glycemic control at least in the studied population of Turkeys ethnicity. However, further clinical studies are warranted to establish solid evidence of the link between MG53 insulin resistance and glycemic control in a wider population elsewhere in the world.”

On page 2, lines 41-42: The author should consider omitting the sentence since the study was not aimed at evaluating the investigations of therapeutics targeting MG53.

Exclusion Criteria and study limitations

On page 5 lines 91-98: The exclusion criteria for insulin resistance was not exhaustive hence posing a dilemma in the conclusion of the study finding. Triglycerides level is one of the significant markers of insulin resistance and beta cell functional reserve. Furthermore, the authors did not use the HOMA-IR assay to screen for insulin resistance. In Table 3: The correlation between MG53 and Triglyceride was significantly positively weak in both groups and significantly positively weak for HDL-C only in the control groups. Since the two groups had significant Triglyceride levels correlated with MG53, it can imply that the control group might have individuals with insulin resistance as well.

In this regard, I suggest the authors expand the study limitations by considering existing uncertainty in the comparison findings between the two groups based on the lack of appropriate exclusion of insulin resistance.

Reviewer #2: The manuscript is well-written and scientifically sound. The authors investigated the association of MG53 with presence of type 2 diabetes mellitus, glycemic control, and diabetic complications. The study is novel and add to the literature of chronic diseases.

6. PLOS authors have the option to publish the peer review history of their article (what does this mean?). If published, this will include your full peer review and any attached files.

Reviewer #1: **Yes: **Festo K. Shayo

Reviewer #2: **Yes: **Doaa Attia

---

## [Author Response · Author response to Decision Letter 0]

16 Jun 2023

June 16, 2023

Dear Editor and Reviewers,

We would like to thank you for your comments, suggestions, and effort in the article. Your correction suggestions have been thoroughly reviewed by all our authors. We would like to state that we agree with your suggestions and criticisms and that your suggested revisions add value to our article. Necessary arrangements have been made in this direction. The revisions made are detailed below, and the current version of the article has been uploaded to the system.

Modifications on reviewer 1's specific recommendations:

1- It has been added to the discussion in detail that the study findings cannot be universally generalized because the study design and population did not meet the external validity, only included participants of Turkish ethnic origin, and insulin resistance in the control group could not be excluded entirely. 

On page 18, lines 272-280: ‘’ When interpreting the results of our study, it should be considered that it was single-centered and performed only in the Turkish ethnic group. There may be an association between MG53 and insulin resistance and diabetes in other ethnic populations elsewhere. Furthermore, given the high prevalence of insulin resistance in Turkey and the lack of comprehensive exclusion of insulin resistance in the studied population, it is challenging to generalize the results of this study. Moreover, there are limited clinical studies and, in particular, multicenter clinical randomized trials or systematic reviews to build solid evidence for this conclusion. The imperative finding of this study cannot be generalized to other populations because the study methods (design, setting, and population) do not meet external validity. The design of our study cannot refute previous evidence but rather provides clinical insight.’’

2- The conclusion part has been updated in line with your suggestion.

On page 18-19, lines 283-286: ‘’ Our research has shown that there is no significant connection between the levels of circulating MG53 and T2 DM or glycemic control in the studied population of Turkish ethnicity. However, we believe that further clinical studies should be conducted to establish a more solid link between MG53, insulin resistance, and glycemic control across a broader range of populations worldwide.’’

3- In the abstract, the conclusion part has been rewritten as you stated, with the term " therapeutics targeting MG53" removed.

On page 2, lines 40-43: ‘’ Our research finding provides insightful clinical evidence of lack of association between the levels of MG53 and T2 DM or glycemic control, at least in the studied population of Turkeys ethnicity. However, further clinical studies are warranted to establish solid evidence of the link between MG53 insulin resistance and glycemic control in a wider population elsewhere in the world.’’ 

4- Your suggestions regarding Exclusion Criteria and study limitations have been detailed in the limitations section and have been rearranged.

On page 19, lines 293-295:’’ Finally, the control group consisted of healthy individuals who did not meet the clinical and demographic parameters associated with insulin resistance. However, the lack of HOMA-IR index data may suggest that the presence of insulin resistance in the control group could not be excluded entirely.’’

Other editorial edits:

1- The article has been checked for compliance with the style requirements of PLOS ONE, including for file naming.

2- Acknowledgments Section has been rearranged as suggested.

On page 19, lines 296-297: ‘’ We would like to express our deep and sincere gratitude to our research supervisor, Dr. S. Guldiken, Endocrinology and Metabolism.’’

3- The full ethics statement has been added to the "Methods" section of our article file.

On page 4, lines 83-84: ‘’ The Ethics Committee of Trakya University approval was granted before the study (TUTF-BAEK 2021/414).’’

4- The reference list has been reviewed to ensure it is complete and accurate.

Kind Regards,

Burak Andac, MD

---

## [Decision Letter · Decision Letter 1]

29 Aug 2023

Association of MG53 with presence of type 2 diabetes mellitus, glycemic control, and diabetic complications

PONE-D-23-16760R1

Dear Dr. Burak,

We’re pleased to inform you that your manuscript has been judged scientifically suitable for publication and will be formally accepted for publication once it meets all outstanding technical requirements.

Kind regards,

SARBASHRI BANK, PhD

Academic Editor

PLOS ONE

Additional Editor Comments (optional):

1) Authors should mention the ethical properly according to journal guide line. 

2) Authors directly explained there is no link between MG53 protein and T2D in a particular ethnicity. The found result would be the same in other population? should explain the main fact.

Reviewers' comments:

Reviewer's Responses to Questions

**Comments to the Author**

1. If the authors have adequately addressed your comments raised in a previous round of review and you feel that this manuscript is now acceptable for publication, you may indicate that here to bypass the “Comments to the Author” section, enter your conflict of interest statement in the “Confidential to Editor” section, and submit your "Accept" recommendation.

Reviewer #1: All comments have been addressed

Reviewer #2: All comments have been addressed

2. Is the manuscript technically sound, and do the data support the conclusions?

Reviewer #1: (No Response)

Reviewer #2: Yes

3. Has the statistical analysis been performed appropriately and rigorously? 

Reviewer #1: (No Response)

Reviewer #2: Yes

4. Have the authors made all data underlying the findings in their manuscript fully available?

Reviewer #1: (No Response)

Reviewer #2: Yes

5. Is the manuscript presented in an intelligible fashion and written in standard English?

Reviewer #1: (No Response)

Reviewer #2: Yes

6. Review Comments to the Author

Reviewer #1: (No Response)

Reviewer #2: All reviewers' comments are addressed. The manuscript is well-written and scientifically-sound and it is suitable for publications at PLoS One

7. PLOS authors have the option to publish the peer review history of their article (what does this mean?). If published, this will include your full peer review and any attached files.

Reviewer #1: **Yes: **Festo K. Shayo

Reviewer #2: No

---

## [Editor Report · Acceptance letter]

4 Sep 2023

PONE-D-23-16760R1 

Association of MG53 with presence of type 2 diabetes mellitus, glycemic control, and diabetic complications

Dear Dr. Andaç:

I'm pleased to inform you that your manuscript has been deemed suitable for publication in PLOS ONE. Congratulations! Your manuscript is now with our production department. 

Kind regards, 

on behalf of

Dr SARBASHRI BANK 

Academic Editor

PLOS ONE